# Structural Basis of Mutation-Dependent p53 Tetramerization Deficiency

**DOI:** 10.3390/ijms23147960

**Published:** 2022-07-19

**Authors:** Marta Rigoli, Giovanni Spagnolli, Giulia Lorengo, Paola Monti, Raffaello Potestio, Emiliano Biasini, Alberto Inga

**Affiliations:** 1Department CIBIO, University of Trento, Via Sommarive 9, 38121 Trento, Italy; marta.rigoli@unitn.it (M.R.); giulia.lorengo@studenti.unitn.it (G.L.); emiliano.biasini@unitn.it (E.B.); 2Mutagenesis and Cancer Prevention Unit, IRCCS Ospedale Policlinico San Martino, L.go R. Benzi, 10, 16132 Genoa, Italy; paola.monti@hsanmartino.it; 3Department of Physics, University of Trento, Via Sommarive 9, 38121 Trento, Italy; raffaello.potestio@unitn.it

**Keywords:** TP53, molecular dynamics, R337H, tetramerization domain, transactivation assays

## Abstract

The formation of a tetrameric assembly is essential for the ability of the tumor suppressor protein p53 to act as a transcription factor. Such a quaternary conformation is driven by a specific tetramerization domain, separated from the central DNA-binding domain by a flexible linker. Despite the distance, functional crosstalk between the two domains has been reported. This phenomenon can explain the pathogenicity of some inherited or somatically acquired mutations in the tetramerization domain, including the widespread R337H missense mutation present in the population in south Brazil. In this work, we combined computational predictions through extended all-atom molecular dynamics simulations with functional assays in a genetically defined yeast-based model system to reveal structural features of p53 tetramerization domains and their transactivation capacity and specificity. In addition to the germline and cancer-associated R337H and R337C, other rationally designed missense mutations targeting a significant salt-bridge interaction that stabilizes the p53 tetramerization domain were studied (i.e., R337D, D352R, and the double-mutation R337D plus D352R). The simulations revealed a destabilizing effect of the pathogenic mutations within the p53 tetramerization domain and highlighted the importance of electrostatic interactions between residues 337 and 352. The transactivation assay, performed in yeast by tuning the expression of wild-type and mutant p53 proteins, revealed that p53 tetramerization mutations could decrease the transactivation potential and alter transactivation specificity, in particular by better tolerating negative features in weak DNA-binding sites. These results establish the effect of naturally occurring variations at positions 337 and 352 on p53’s conformational stability and function.

## 1. Introduction

The *TP53* gene encoding the well-known tumor suppressor protein, p53, is undoubtedly one of the most critical cancer genes [1]. Acting primarily as a nuclear sequence-specific transcription factor, p53 coordinates a complex network of gene targets to modulate many cellular pathways that respond to many cellular stress conditions [2]. The p53 transcriptional network is pleiotropic. The responses are distributed to many distinct target genes that collectively determine a specific cell outcome among the many possible, including cell cycle arrest and cell death as archetypes [3]. Furthermore, the p53-controlled pathway is highly regulated by multiple feedback loops at transcriptional, post-transcriptional, and translational levels [1,4]. This complexity is exemplified by the notion that there is still a significant lack of knowledge regarding the critical tumor-suppressive functions of p53 in specific cancer types or evolutionary cancer tracks [5,6,7].

The p53 function as a transcription factor relies on the assembly of a p53 tetramer that can interact with variations of DNA target sites known and referred to as response elements (REs) [8,9,10]. The p53 tetramer conformation results in a dimer of dimers, with p53 dimers being formed co-translationally [11]. This assembly is made possible by the presence and features of a so-called oligomerization or tetramerization (TET) domain in the proximal carboxy-terminal region (C-ter) of the protein [10,12]. The structure of the human p53 TET domain has been resolved by crystallography and NMR and consists of a four-helix bundle, with each p53 monomer featuring a sheet-loop-helix fold [13,14]. p53 tetramer assembly and stability are also influenced by interactions between the DNA-binding domains (DBDs). These are thought to be modulated by long-range interactions between the distal carboxy-terminal region of the protein and the DBDs [14,15,16]. The transactivation domains are critical for p53 function [6,17] and are located in the amino-terminal sequence (within the first 60 amino acids), a region that is considered to be intrinsically disordered [18,19,20,21]. Together with the last 30 amino acids in the C-ter, it is subject to several post-translational modifications that impart changes in protein–protein interactions, conformation, subcellular localization, and function to the protein [22,23,24]. p53 post-translational modifications were also shown to affect long-range intramolecular interactions between either the N-ter or the C-ter and the DBDs, influencing DNA binding [20,21].

The p53 binding site in the DNA reflects and co-evolved with the quaternary conformation of the protein. As the active protein structure is a p53 tetramer, so the p53 REs comprise an arrangement of four binding sites, consisting of five nucleotides, which are organized as inverted repeat dimers (i.e., a half-site), and two such dimers are placed adjacent or closely spaced to form an entire site [25,26,27,28,29]. Many studies based both on in vivo approaches, such as ChIP-seq or ChIP-Exo, and in vitro assays, such as Selex-seq, have derived the features of the p53 consensus RE and cataloged the many variants present in genomes (reviewed in [28,30,31]). Further, biochemical, biophysical, or defined gene reporter assays have deconstructed the p53 RE to reveal its critical features [25,27,31]. These studies also established the consequences of polymorphisms or mutations in the p53 REs or the p53 protein itself, which are among the most frequent somatic alterations acquired by human cancer [32,33,34,35,36]. Studies also established that p53 tetramers could be assembled on DNA as dimers, and p53 tetramers can also bind hemi-specifically, with one dimer establishing sequence-specific contacts with a decamer binding site and the second p53 dimers contacting non-specifically the DNA backbone [12].

Interestingly, there is clear evidence not only that the p53 RE is highly degenerated, which results in many different versions of the binding site, enabling various levels of quantitative controls by p53, but also there has been an apparent selection pressure during evolution to avoid optimal p53 binding sites in promoters that would lead to constitutive, non-regulatable p53-dependent transcriptional control of a target gene [25,36,37].

Given the essential requirement for the tetrameric structure, it is not surprising that the TET domain has undergone a significant evolutionary divergence within the p53 family of proteins that also comprises p63 and p73 [38,39,40,41]. In addition, within p53 protein from different and distant species, there is evidence of a divergence in the features of the TET domain that can also impact how the p53 tetramer interacts with the DNA, particularly for what concerns the relative arrangements of the two DNA half-sites [42,43]. 

Within the spectrum of p53 mutations observed in human cancers, those at the TET domain are much less abundant than mutations in the central DBDs of the protein; in fact, nearly 80% occur at residues in this latter domain [44]. p53 mutations are among the most frequent somatic alterations occurring in human cancer [1]; furthermore, p53 mutations can also be inherited in the germline, where heterozygous mutant alleles are associated with a highly penetrant cancer predisposition syndrome [45,46]. Moreover, in the case of germline p53 alleles, there is a strong enrichment for missense mutants in the p53 DBDs [47].

The low frequency of mutations in the p53 TET domain can be explained, at least in part, by the fact that p53 missense mutations in the DBDs can also acquire oncogenic gain of function (GoF) [48,49]. GoF, however, requires that the mutant protein remains able to be localized in the nucleus and participate in protein complexes, modulating aspects of DNA replication or transcription. Conversely, an intact P53 tetrameric conformation is critical for masking a nuclear export signal. However, tetramerization can enable the occurrence of mixed tetramers comprising wild-type (WT) and mutant DBDs in cells undergoing a transformation that have acquired a heterozygous p53 mutation, a feature of mutated alleles frequently referred to as dominant-negative [50,51]. Finally, systematic mutagenesis of the human p53 TET domain based on single-nucleotide changes showed that most of those mutants would not dramatically reduce the p53 transactivation function in a reporter assay [36]. 

There are, however, a few exceptions to the general rule of the scarcity of p53 TET domain mutations in cancer [52]. Two missense changes in the p53 TET domain have been found in the germline (R337H and R337C), affecting a conserved arginine involved in establishing a salt-bridge with D352, which is important for the stability of the dimeric conformation (Figure 1). While germline R337C is associated with a classical tumor proneness spectrum, R337H was initially identified as predisposing to pediatric adrenocortical carcinoma [53,54]. Molecular epidemiology studies, a screening campaign, and a knock-in mouse model have clarified that this allele results in a partially functional protein, whose degree of inactivation could be related to pH and the protonation level of the histidine residue [55,56,57]. The allele is widespread in the population of southern Brazil. It is associated with variable penetrance to various pediatric and adult-onset cancers, including choroid plexus carcinomas and breast cancer [54]. The arginine 333, 335, and 337 were found to be post-translationally modified by PRMT5, leading to either mono- or di-methylation [58]. Based on a triple arginine to lysine mutant and on the impact of PRMT5 silencing in osteosarcoma-derived cells, it was inferred that arginine methylation in the p53 TET domain can impact transactivation specificity [58]. 

Given the centrality of p53 in cell biology, the strong selection to lose or at least attenuate p53 responses in cancer and the fact that mutant p53 proteins are overexpressed in cancer cells, the potential to reactivate p53 mutant proteins by pharmacological strategies as cancer therapy has a robust rationale and has been pursued in many ways [59,60,61]. Cell-based assays have identified potential small molecules of therapeutic interest, but it has proven difficult to establish the exact mechanism of action of the hits coming from these assays [62]. Potent rationally designed molecules have also been developed primarily targeting the interaction between p53 and its major negative regulator MDM2 but have shown high toxicity in clinical trials [63]. Molecules patching the thermodynamic instability caused by specific p53 mutations, such as Y220C, or trying to increase the stability of the p53 DBDs have also provided proof of concept evidence of the value of rescuing p53 folding [64]. The lack of a high-resolution structure of the full-length p53 tetramer bound to DNA represents a limitation in rational drug design. With a few exceptions, all the effort has been directed toward p53 mutant proteins in the DBDs [65,66,67]. Molecules targeting the TET domain could be highly valuable, especially for individuals who have inherited the R337H mutation [68] but, in principle, also for other classes of cancer-associated p53 alleles in the DBDs that retain partial function and which have a defect that could, in principle, be compensated for by increasing the stability of the tetramer conformation. 

In silico modeling, particularly molecular dynamics (MD), has been attempted to boost the search for small molecule leads to rescue p53 mutations [69]. Still, until recently, those approaches have been somewhat limited by the relatively short simulation times that could be afforded.

To our knowledge, a single systematic computational study on the stability of the TET domain has so far been conducted by Lwin et al. [70]. Here, the authors investigated by means of all-atom MD simulations the WT p53 TET domain as well as the R337H mutant at different salt concentrations and pH conditions. MD simulations of various configurations of the TET domain were performed in explicit and implicit solvation, for a total of 130 ns subdivided into eight different trajectories, coherently with the standards of the times. In the aforementioned work, the existence of a fluid salt-bridging cluster composed of residues R337, D352, R333, and E349 as well as the destabilization of the WT protein and of the R337H mutant in acidic conditions was pointed out. No differences in the remaining six simulated systems were observed in terms of stability of the protein structure (RMSD), in particular the R337H systems were found to be comparable in terms of RMSD to the WT ones. 

Here, we adopted a state-of-the-art MD approach to study the p53 TET mutant proteins R337C and R337H along with rationally designed mutations to probe the impact of the salt-bridge between R337 and D352 on the stability of the p53 TET domain and the consequences not only on the transactivation potential but also on transactivation specificity. Indeed, by coupling computational predictions through extended all-atom molecular dynamics simulations with the results of a genetically defined transactivation-based assay in yeast, we reveal that targeting the R377-D352 salt-bridge interaction in the p53 TET domain can lead to reactivation of p53 but also an apparent change in relative transactivation specificity.

## 2. Results

### 2.1. Computational Prediction of the Effect of Naturally Occurring or Rationally Designed TP53 Mutation on the Stability of the p53 TET Domain

Mutations within the TET domain of p53 have been associated with Li-Fraumeni or Li-Fraumeni-like cancer predisposition syndromes. These mutations are thought to affect the stability of the tetramer by altering the intermolecular salt-bridge formed by residues R337 and D352 (Figure 1). We sought to test this possibility by performing extended, all-atom molecular dynamics simulations. First, we simulated for 2 µs the WT p53 TET domain and the two disease-associated mutant proteins (i.e., R337C and R337H). We included three different states of His337, accounting for the protonation of δ (R337H_δ_), ε (R337H_ε_), or both (R337H_δε_) imidazole nitrogens. 

Next, we simulated for 2 µs the three artificial p53 mutant proteins, designed to test the contribution of the intermolecular electrostatic interaction between residues 337 and 352 (i.e., D352R, R337D, and R337D/D352R). Our simulations indicated that disease-associated substitutions at residues 337 (i.e., R337H_δ_, R337H_ε_, and R337C) destabilize the conformation of the p53 TET domain. Interestingly, the R337H_δε_ form_,_ which carries a positive charge on the side chain similar to its WT counterpart, showed no stability alterations (Figure 2a and Appendix A). Artificial p53 mutant proteins showed differential destabilization patterns, with the R337D exhibiting the most prominent alteration, the R337D/D352R a more modest effect, while the D352R resembled the WT form (Figure 2b and Appendix A). 

To corroborate these data, we computed the interchain distances of residues 337 and 352 within the p53 tetramer for all of the simulated systems. The results showed that similarly to the WT, a charged histidine at position 337 and the double-mutant protein R337D/D352R displayed distance distributions compatible with the formation of stable salt-bridges. Close interaction distances were also observed for residues 377 and 352 in the D352R mutant, possibly reflecting π-π and π-cation interactions between the guanidinium groups of the arginines. Conversely, in all of the other conditions, a distance incompatible with a stable interaction was detected in at least two pairs of residues (Figure 3).

The alteration of an intermolecular interaction resulting from an amino acid substitution may destabilize regions in the protein complex several residues apart. To assess this possibility for residues 337 and 352 of the p53 TET domain, we computed the average contact map differences between the trajectories of the WT and the mutant forms. As expected, the contact maps of D352R and R337H_δε_ displayed no substantial deviation from the WT. Conversely, R337C, R337D/D352R, R337D, R337H_ε_, and R337H_δ_ showed a significant variation, with the latter presenting the most prominent effects (Figure 4 and Appendix A). 

In summary, these results define the mechanism by which residue variations in positions 337 and 352 destabilize the TET domain of p53 and predict a structure–function relationship.

### 2.2. Experimental Analysis of Mutation-Dependent, p53-Driven Transcription

Based on the results of the MD simulations, we designed a set of experiments to validate the predicted effects of each p53 mutant protein. The goal was achieved by taking advantage of a defined functional assay in yeast, where p53 alleles can be expressed at variable levels under an inducible, finely tunable promoter, and the p53 transactivation potential is quantified based on the level of activation of the firefly reporter gene that is cloned in a single copy at a specific chromosomal location. A panel of reporter strains that are entirely isogenic, except for the sequence of the p53 RE driving p53-dependent reporter gene transactivation, were used to establish the consequences of p53 alleles on sequence-specific transactivation specificity. Specifically, we tested WT p53, the naturally occurring R337C and R337H alleles, and the *ad hoc* designed mutations R337D, D352R, and the double-mutant R337D/D352R. The relative transactivation of each protein was determined by culturing yeast reporter strains stably transformed with centromeric p53 reporter plasmids and cultured in media containing different amounts of galactose to vary the level of expression of p53 proteins. The results confirmed that R337C is a near loss-of-function mutation. At the same time, R337H exhibited a more subtle transactivation defect that is appreciable, particularly when p53 protein levels are low Figure 5 and Appendix A), consistent with previous studies. R337D is a complete loss of function allele, while D352R is WT-like. The double-mutant R337D/D352R showed only a partial rescue of the R337D effect (Figure 5 and Appendix A). 

Interestingly, we noticed that the relative activity of this panel of mutations was affected by the level of expression of the p53 alleles and by the nature of the p53 RE. We tested two REs derived from the very well-established p21 target gene; one RE was the natural sequence derived from the human promoter and known as p21-5′, while the second was derived from p21-5′ but contained a two-nucleotide spacer between the decameric half-sites (p21-SP2). Consistent with previous studies, we confirmed that the small spacer significantly diminished the p53 transactivation potential and required higher galactose levels to measure p53-dependent transactivation (Figure 5 and Appendix A). Unexpectedly, however, the presence of the spacer also led to a change in the relative activity of the p53 TET domain. In particular, D352R showed much higher transactivation potential than WT p53, and, consistent with that observation, the double-mutant R337D/D352R was fully rescued. Moreover, the R337H allele was WT-like in the p21-SP2 reporter strain or slightly more active. These differences in relative activity were not solely related to the lower DNA-binding affinity of WT p53 for the REs. We also tested the panel of TET mutations with one additional yeast reporter strain based on the p53 RE found in the human PUMA/BBC3 gene. This RE has no spacer between the decameric half-sites, like the canonical p21-5′ RE, but, due to the presence of deviations from the RE consensus sequence, it mediates moderate responsiveness to p53. R337C was confirmed as a near loss-of-function allele in the PUMA reporter strain, R337D as a complete loss of function, and R337H and D352R as partial function alleles. The double-mutant R337D/D352R was only slightly active (Appendix A). Finally, when p53 expression levels were high in 0.032% galactose (or 0.128% for the p21-SP2 reporter), R337H and D352R were WT-like, and the double mutant was fully or nearly fully rescued. Still, only the structurally altered p21-SP2 reporter showed enhanced activity of D352R. All of the above-described differences in transactivation potential were undoubtedly not related to variations in the level of protein expression, as confirmed by Western blot (Appendix A).

## 3. Discussion

High-resolution computational modeling that assesses the stability and dynamic folding changes in the p53 protein will be essential not only to reveal the molecular basis of functional alterations caused by disease-associated mutations but, in principle, also to identify structural intermediates that could be qualified as target sites for the design of pharmacological rescue molecules. In the case of p53, the possibility of developing drugs that can restore its conformation or improve its thermodynamic stability would have high potential in benefitting cancer patients [59,60,61,63,71,72]. Drugging p53 has, however, proven very difficult, not only because the protein is a transcription factor and, hence, lacks a well-defined catalytic site but also because of the incomplete availability of data on the structure and dynamics of the entire functional unit of the protein, i.e., a p53 tetramer that is competent to bind to DNA and stimulate transcription. There are crystal structures available for the p53 DNA-binding domain, both the WT, alone or bound to different DNA target sites, some hotspot p53 mutations [10,73,74,75,76,77,78,79,80], and the p53 TET domain [13]. However, there is still a lack of data to fully reveal the structural details of the crosstalk between the different domains, including the intrinsically disordered and heavily post-translationally modified N-ter and C-ter domains [10,16,19,24,81,82]. This lack of knowledge represents a severe limitation, as p53 is a highly versatile transcription factor that interacts through a wide range of affinity with many DNA RE-binding sites that can be structurally diverse in their internal organization [12,29]. Hence, while increasing the thermodynamic stability of the p53 DNA-binding domain can increase p53 transactivation and rescue the consequences of some cancer-associated mutations [83,84,85,86], it can also affect transactivation specificity towards the various categories of p53-binding sites [1,3,30], with functional effects that are difficult to predict [87].

Conversely, cell-based assays can select for small molecules that result in a specific p53-dependent cellular outcome [61,63,88]. Still, given the extreme pleiotropy of p53 and the pluralism of highly connected molecular pathways, it has proven challenging to reveal the precise mechanisms of action on p53 of leads emerging from those screening campaigns. All-atom MD simulations hold great potential for revealing the molecular defect of p53 mutations in terms of local structural distortions, long-range spatial effect, and kinetic consequences on conformational changes [69,89,90,91]. MD has been applied to study a panel of hotspot p53 missense mutations in the DBD [92] to model the intrinsically disordered p53 transactivation domain to empower the screening of binding compounds [19] and to study the stability of the p53 TET domain [70]. There have also been attempts to combine high- with low-resolution modeling to address the entire p53 tetramer [89,93]. For example, those studies have suggested the existence of specific structural features that are more evident or stable in the mutant p53 DBDs and could be druggable [92]. In the case of the p53 TET domain, structure and molecular dynamics data have led to the attempt to design molecules that could lead to enhanced stability [68]. Our results directly suggest that the TET domain and DNA-binding domain are not independent, as an alteration in the TET domain could also impact the arrangements of p53 monomers and dimers which, in turn, results in a variety of functional consequences. This finding is consistent with recent data on the existence of complex intramolecular interactions between p53 domains and on the impact that post-translational modifications can have on those interactions, indirectly affecting DNA-binding affinity, specificity, or the balance between sequence-specific and nonspecific DNA binding [19,20,21,58]. The result that the D352R rescues the loss of function R337D mutant and exhibits wild-type or even higher transactivation is, on the one hand, consistent with the important role of the D352:R337 salt-bridge interaction but also suggest a more dynamic interplay between different arginine residues in the p53 TET domain [70]. Importantly, post-translational arginine-methylation in the p53 TET domain can impact on p53 function as a transcription factor [58]. Overall, our findings confirm that only the combination of structural and modeling studies, along with cell-based assays monitoring p53 function, can fully reveal the dynamic structure/functional features of the full-length p53. Such information could allow future therapeutic strategies to restore mutant p53 stability in human cancer.

## 4. Materials and Methods

### 4.1. Molecular Dynamics Simulations

Starting from the WT PDB structure (PDB code: 2J0Z), the mutant models were built using the UCSF Chimera package developed by the UCSF Resource for Biocomputing, Visualization, and Informatics [94]. The proteins were solvated in water using the TIP3P model [95]; Na and Cl ions were added to neutralize the protein’s charge and mimic the physiological salt concentration (150 mM). The simulation box was chosen for the cubic shape, and the protein had a minimum distance of 1 nm to the box’s edge. The simulations were performed in the NPT ensemble at 300 K and 1 bar, using the stochastic velocity-rescale thermostat [96] and the Parrinello–Rahman barostat [97]. The chosen parameters were temperature coupling constant tau-t = 0.1 ps and pressure coupling constant tau-*p* = 2 ps. The integration step was set to 2 fs, the selected integrator was the leap-frog one, and the LINCS algorithm [98] was selected to apply holonomic constraints. The forcefield employed for the protein simulations was the Amber14sb [99]. MD simulations and RMSD analyses were performed using the Gromacs 2018 software [100,101]. The contact map calculations were done by means of an in-house Python script following the definition provided by Bonomi et al., [102].

### 4.2. Yeast Cultures

The yeast reporter strains were previously developed from the yLFM-CORE strain and the so-called *Delitto Perfetto* targeting strategy [103]. Cells were transformed with centromeric plasmids containing the expression cassette for human p53 alleles [32]. The promoter was derived from the *GAL1,10* gene, and we established it could be regulated by varying the sugar concentrations in the culture medium. Specifically, the expression of p53 was repressed in glucose media, reaching a slightly higher basal level when glucose was replaced by galactose, and it could then be gradually induced to moderate or high levels by adding galactose to the raffinose-containing medium [104,105]. Transformants were selected based on the TRP1 marker present in the plasmid and were kept in glucose media until the day of the transactivation experiment. Five independent transformants were tested for each p53 mutant protein by preparing fresh patches on selective glucose plates to start liquid cultures for the luciferase assays.

### 4.3. Luciferase Assay

To measure the p53-dependent transactivation of the firefly reporter, we used an optimized low-volume liquid culture system [43,105]. Briefly, independent transformants were resuspended from patches on glucose plates and placed in 200 μL of selective liquid medium containing 2% raffinose as a carbon source within a 96-well plate. Sixty microliters of cell suspension were then transferred to different 96-well plates and mixed with an equal volume of selective medium containing raffinose as a carbon source, supplemented or not with a desired amount of galactose to induce at different levels the expression of p53. Plates were incubated at 30 °C with moderate shaking for 6 h. In previous experiments, we determined that between 6 and 8 h in this culture condition, the reporter’s expression of p53 and p53-dependent transactivation reached a peak [25]. To measure the luciferase activity, 10 μL from each well of the cultures were transferred to a white 384-well plate. An equal volume of 2× PLB buffer (Promega, Promega Italia, Milan, Italy) was added to permeabilize the yeast cells. Permeabilization was achieved during 10 min of incubation in a shaker at room temperature. In the meantime, the optical density at 600 nm of each culture was measured in the 96-well plate using a plate reader. Luciferase activity was quantified by adding 10 μL of the substrate (Promega) to the permeabilized culture samples in the 384-well plate and measuring the luminescence in a plate reader. Luciferase activity was normalized relative to the optical density of the cultures. As a control, transformants with an empty plasmid that did not express p53 were processed in the same manner, and the low level of p53-independent expression of the firefly reporter was subtracted from the other values.

### 4.4. Western Blotting

To compare p53 protein expression, we developed 3 mL liquid cultures in falcon tubes using a selective medium and conditions equivalent to the luciferase assay experiment. Two microliters of the cultures were collected in 2 mL Eppendorf tubes after six hours and centrifuged at 4000 rpm for less than 2 min to obtain a pellet. Cells were washed once with 5 mL of sterile water and centrifuged again to obtain a pellet that was then resuspended in 150 μL of 2× PLB buffer. We then added the approximate equivalent of 150 μL of acid-washed, sterile glass beads. The tubes were vortexed at maximum speed for cycles of 30 s, followed by one-minute of rest on ice. After six such cycles, the tubes were spun at maximum speed in a refrigerated microfuge at 4 °C for 15 min. The supernatant was then transferred to a new tube, trying to maintain the solution cold. Proteins were quantified using the bicinchoninic acid assay (Pierce, Thermo Fisher Scientific, Milan Italy). Western blot was performed with a standard SDS-PAGE protocol in 10% acrylamide gels and loading of 10 μg of total protein for each sample. Immunodetection was performed after transferring the protein onto a nitrocellulose membrane, using the DO1 antibody against p53 and an anti PGK1 antibody to detect a reference protein and revealed using a secondary antibody and the ECL detection kit (Amersham, Fisher Scientific, Milan, Italy). Band intensities were quantified using ImageJ.

## Figures and Tables

**Figure 1 ijms-23-07960-f001:**
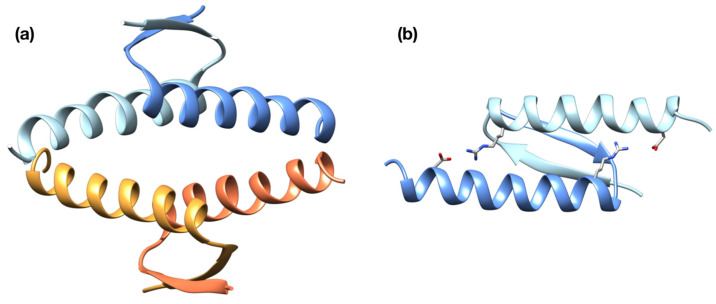
**Structure of the p53 TET domain:** (**a**) ribbon representation of the fully assembled TET domain of WT p53, where the individual chains are depicted in different colors; (**b**) representation of a dimer of the p53 TET domain displaying the two intermolecular salt-bridges between residues R337 and D352.

**Figure 2 ijms-23-07960-f002:**
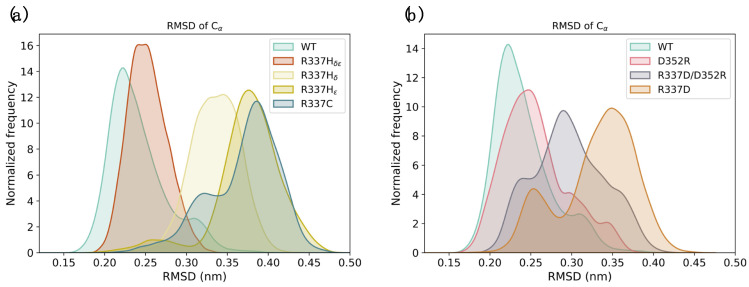
**Molecular dynamics simulations of p53 TET domain mutants.** (**a**) The destabilization effect of disease-associated mutations lying within the p53 TET domain was analyzed by molecular dynamics. The graph shows the RMSD distribution of the WT tetramer and the R337C and R337H mutants. The latter was simulated in three different states of the histidine including the protonation of δ (R337H_δ_), ε (R337H_ε_), or both (R337H_δε_) nitrogens. The results indicate that the WT and R337H_δε_ showed overlapping distributions with low RMSD values. Conversely, the R337H_δ_, R337H_ε_, and R337C mutants exhibited distributions centered at higher RMSD values. (**b**) The effect of rationally designed substitutions, including D352R, R337D, and R337D/D352R, on the stability of the p53 TET domain was compared to the WT counterpart. The results indicate that the WT and the D352R form showed distributions with low RMSD values. Conversely, the R337D/D352R and R337D exhibited distributions centered at higher RMSD values, with the latter presenting a more prominent destabilization. Each system was simulated for 2 µs.

**Figure 3 ijms-23-07960-f003:**
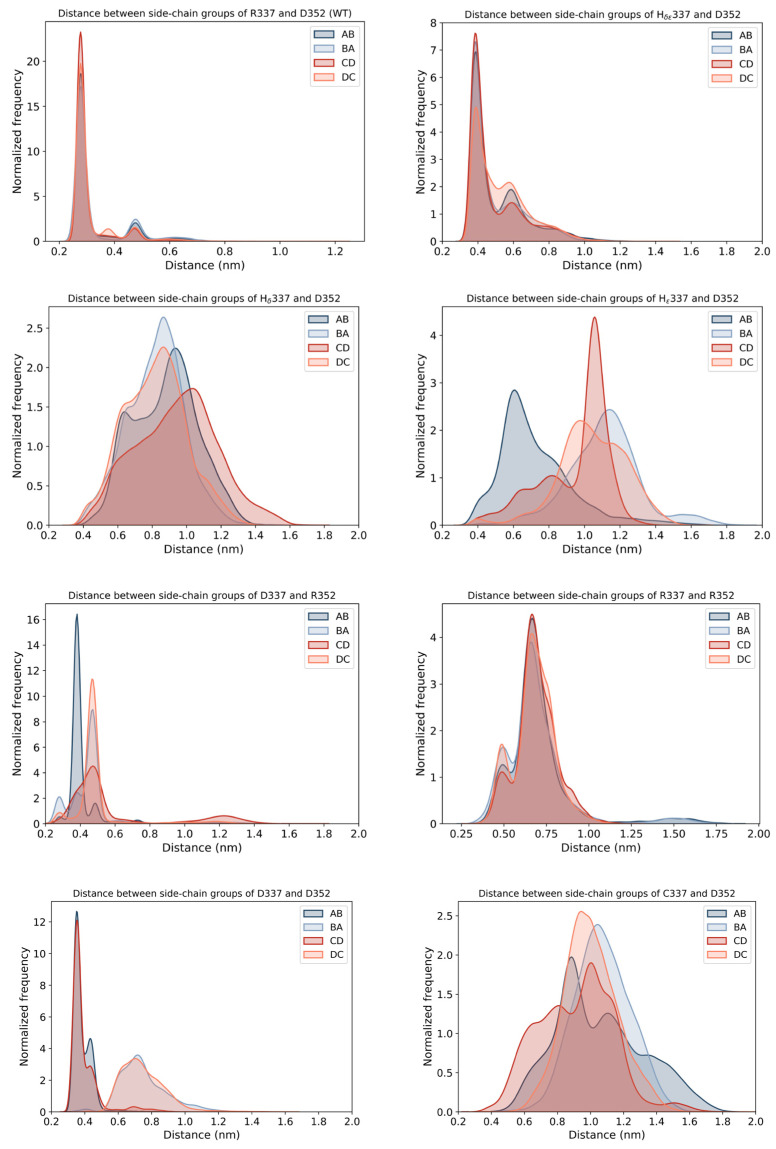
**Analysis of the distances between side chains of residues 337 and 352 within the p53 tetramer.** The interchain distance between residues involved in the salt-bridge formation at positions 337 and 352 in the WT protein was analyzed for all of the simulated systems. The graphs display the measure of the distance between the centers of mass of the functional groups of each couple of amino acids (337 and 352), initially forming salt-bridges between two pairs of the four polypeptide chains of the tetramer (A with B and C with D), resulting in the 4 different arrangements (i.e., 337.A-352.B, 337.B-352.A, 337.C-352.D, and 337.D-352.C). The results show that only the charged histidine at position 337, the double-mutant R337D/D352R, and the D352R mutant displayed distance distributions compatible with the formation of effective interactions.

**Figure 4 ijms-23-07960-f004:**
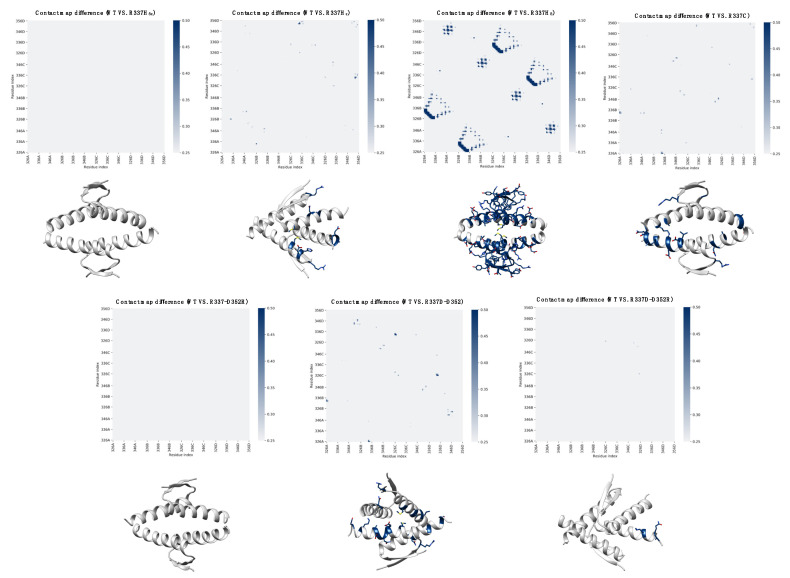
**Contact map analysis of WT and mutant p53 TET domain.** The average contact map difference is represented as a residue index matrix for each form. Mutant p53 forms are expressed as the absolute value of the difference with the WT protein. The contact maps of R337C, R337D/D352R, R337D, R337H_ε_, and R337H_δ_ show a significant variation, with the latter presenting the largest divergence. In contrast, D352R and R337H_δε_ displayed no substantial deviation from the WT. The side chains of residues with the most evident differences in each contact map, defined by a threshold between 0.25 and 0.5, are illustrated as sticks.

**Figure 5 ijms-23-07960-f005:**
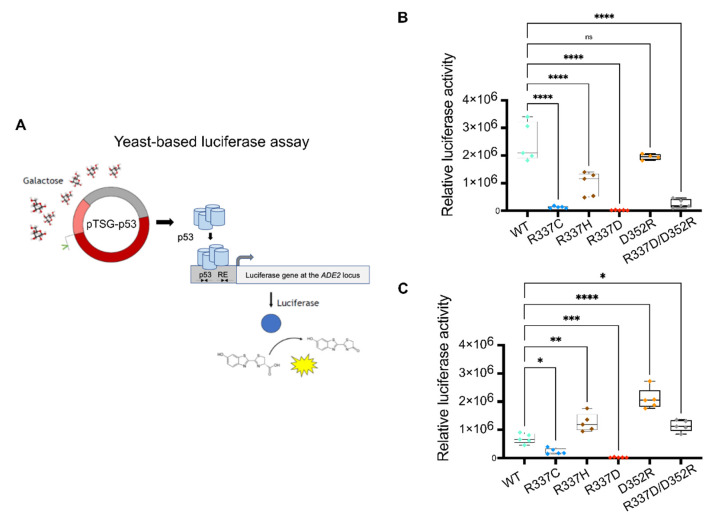
**Transactivation potential of p53 TET mutations.** (**A**). Scheme of the experimental approach. p53 expression was achieved by stable transformation of yeast reporter cells with plasmids that, by containing an origin of replication and a centromeric sequence, ensured stable low-copy transmission to daughter cells. The vector was based on the pRS314 shuttle vector and contained a selectable TRP1 marker gene complementing an auxotrophy of the yeast strain. The vector contained a GAL1,10 promoter driving the expression of the human p53 cDNA. This promoter affords inducible expression that can be modulated precisely by varying the amount of galactose in the medium. The yeast reporter strain contained the Photinus Pyralis luciferase cDNA cloned on chromosome XV at the endogenous ADE2 locus. The luciferase transcription was regulated by a minimal promoter that can be stimulated by p53 binding to upstream REs. Permutation of the p53 RE in isogenic yeast strains was made possible by an oligonucleotide targeting approach (see Section 4 for details). (**B**) Relative transactivation of the indicated p53 alleles expressed at moderate levels by medium containing 0.008% galactose for six hours in a yeast reporter strain containing the high-affinity p53 RE derived from the p21 promoter (p21-5′). Luminescence was normalized to the optical density of the cultures. Plots present the average normalized light units, the confidence intervals, and the individual data points after subtracting the p53-independent activity of the reporter. (**C**) Same as in (**B**), except that the experiment was performed using a reporter strain that relied on a p53-binding site derived from the p21-5′ RE but modified by the inclusion of a 2nt spacer between the two decameric half-site motifs (p21-SP2). Given the lower responsiveness of the spaced RE, a 0.032% galactose concentration was used. * *p* < 0.05; ** *p* < 0.01; *** *p* < 0.001; **** *p* < 0.0001; ns = not significant; one-way ANOVA with Dunnett’s multiple comparison test.

## Data Availability

Not applicable.

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
