# Peer review of "Structural Basis of Mutation-Dependent p53 Tetramerization Deficiency"

_ijms, 2022, doi:10.3390/ijms23147960_

Round 1
Reviewer 1 Report
In the manuscript: ‘Structural Basis of Mutation-Dependent p53 Tetramerization Deficiency by Marta Rigoli et al.’, the authors have explored the impact of the salt bridge between R337 and D352 on the stability of the p53 TET domain, by employing Molecular Dynamics MD (for structural evidence) and assays in yeast (for functional evidence, ie activation potential). Their conclusions point towards a functional role of this bridge for the specificity of p53’s transactivation activity. These findings are interesting for understanding how structural characteristics and the crosstalking between the N-terminal and C-terminal domains of p53, alter p53’s activity, which is vital for translational applications.
A few comments of minor significance.
Line 62: More precisely, these regions are considered to be intrinsically disordered, rather unstructured (10.1039/D0SC04670A, 10.1186/s12885-019-6118-y, 10.1073/pnas.1814051115). This intrinsic disorder allows a high regulatory effect by PTMs.
Line 64: There have been some more recent studies exploring how post-translational modifications on the TAD induce changes in protein-protein interactions, conformation, subcellular localization, and p53 activation, like: 10.1093/jmcb/mjab047, 10.1093/jmcb/mjy049, 10.1073/pnas.2021456118 and more. Studies involving experimental evidence (wet assays) confirming in silico findings, should be highlighted.
The authors should put the conclusions of the effect of the salt bridge, into a more general structural context. For example, they could discuss and compare their findings on the tetramerization of p53, with those by Krois et al: 10.1073/pnas.1814051115. It will be interesting to also discuss/compare the methods and (wet lab) techniques that are required to conclusively describe the dynamics of these structures.
Author Response
We thank the reviewer for the positive evaluation and the important comments.
-Line 62 and ~line 339: we replaced “unstructured” with the more accurate “intrinsically disordered as suggested and referenced two of the studies suggested. the more precise definition and referenced the paper.
-Line 64: we thank the reviewer for pointing us to those recent studies. We added a sentence starting in line 63 to highlight those findings.
-We also added a sentence starting on line 125 to reference the study on arginine 337 methylation by PRMT5
-Discussion: we have included a reference to the molecular dynamics study of the p53 transactivation domain and commented more explicitly on the evidence for long range interactions between p53 domains and their impact on DNA binding affinity/specificity.
We also commented the results obtained with the D352R mutant in the context of the salt-bridge interactions and on the necessary combination of experimental and modeling methods to investigate the interplay between p53 domains on p53 functions.
All changes are in red font.
Reviewer 2 Report
The authors analyzed some consequences of naturally occuring cancer-associated and in vitro mutagenesis-generated missense mutations in the tetramerisation domain of the p53 tumor suppressor protein. The structural consequences of the mutations were predicted by computational modeling, while the impact of mutations on transactivation potential and specifity of p53 protein was studied using a sophisticated approach in yeast reporter strains. The study is thoughtfully designed, well-executed and properly presented in this manuscript. In addition, this experimental approach provides a useful model system for the rational development and testing of p53-targeting anticancer drugs.
Author Response
We thank the reviewer for the positive evaluation of our work. We hope that the changes introduced in the revision process (red font) will also be approved.